# An Integrative Bioinformatic Analysis for Keratinase Detection in Marine-Derived *Streptomyces*

**DOI:** 10.3390/md19060286

**Published:** 2021-05-21

**Authors:** Ricardo Valencia, Valentina González, Agustina Undabarrena, Leonardo Zamora-Leiva, Juan A. Ugalde, Beatriz Cámara

**Affiliations:** 1Laboratory of Molecular Microbiology and Environmental Biotechnology, Department of Chemistry and Center for Biotechnology Daniel Alkalay Lowitt, Federico Santa María Technical University, Valparaíso 2340000, Chile; ricardo.valenciaa@alumnos.usm.cl (R.V.); valentina.gonzalez@alumnos.usm.cl (V.G.); agustina.undabarrena@usm.cl (A.U.); leonardo.zamoral@sansano.usm.cl (L.Z.-L.); 2Millennium Initiative for Collaborative Research on Bacterial Resistance (MICROB-R), Santiago 8320000, Chile; juan@ecogenomica.cl

**Keywords:** keratinases, keratinolytic proteases, marine-derived *Streptomyces*, genomic comparison

## Abstract

Keratinases present promising biotechnological applications, due to their ability to degrade keratin. *Streptomyces* appears as one of the main sources of these enzymes, but complete genome sequences of keratinolytic bacteria are still limited. This article reports the complete genomes of three marine-derived streptomycetes that show different levels of feather keratin degradation, with high (strain G11C), low (strain CHD11), and no (strain Vc74B-19) keratinolytic activity. A multi-step bioinformatics approach is described to explore genes encoding putative keratinases in these genomes. Despite their differential keratinolytic activity, multiplatform annotation reveals similar quantities of ORFs encoding putative proteases in strains G11C, CHD11, and Vc74B-19. Comparative genomics classified these putative proteases into 140 orthologous groups and 17 unassigned orthogroup peptidases belonging to strain G11C. Similarity network analysis revealed three network communities of putative peptidases related to known keratinases of the peptidase families S01, S08, and M04. When combined with the prediction of cellular localization and phylogenetic reconstruction, seven putative keratinases from the highly keratinolytic strain *Streptomyces* sp. G11C are identified. To our knowledge, this is the first multi-step bioinformatics analysis that complements comparative genomics with phylogeny and cellular localization prediction, for the prediction of genes encoding putative keratinases in streptomycetes.

## 1. Introduction

Keratin is a fibrous and recalcitrant protein belonging to a large family of structural proteins that constitute hair, wool, feathers, nails, bristles, and horns of several animals [1]. This protein is densely packed in α-helix or β-sheet structures (α/β-keratin, respectively) and their high degree of cross-linkages by disulfide and hydrogen bonds confers high mechanical stability and resistance to common proteolytic enzymes such as pepsin, trypsin, and papain [2]. Its recalcitrant structure is a significant challenge for degradation, a process that naturally can take as long as many years [3]. Indeed, there are many known examples of preserved hair, skin, and feathers on archeological materials [4,5,6]. Its accumulation can be a major problem, especially for poultry processing farms, where feather keratin is a by-product of industrial activities. Therefore, finding microorganisms that present the ability to degrade this protein will become an eco-friendly alternative to improve the management of their waste resources. So far, several microorganisms have been reported to produce keratinases, including Fungi [7,8] and Bacteria, such as *Bacillus* [9,10,11] and *Streptomyces* [12,13,14,15,16].

Keratinases (EC 3.4.21/24/99.11) are a particular class of proteases that possess keratinolytic activity, being able to degrade several insoluble keratin substrates [3]. They are primarily extracellular; however, cell-bound [17] and intracellular [18] enzymes have also been reported. Most of the keratinases known to date are serine proteases [14,16,19], while a few have been classified as metalloproteases [20]. Nevertheless, it remains unclear which features are required for keratin degradation [21]. Most studies are focused on the purification and characterization of a unique keratinase or protease with keratinolytic activity [16,22,23,24,25,26]. However, purified keratinases known to date cannot completely solubilize native keratin, which confirms that one enzyme alone cannot fully decompose its recalcitrant structure [3,27]. For this reason, it has been hypothesized that a pool of proteases is needed to penetrate and break down keratin structure, instead of a single enzyme. Indeed, in natural environments, this degradation could result from the combined action of enzymes from various organisms, such as Fungi and Bacteria [3]. Moreover, due to the high cysteine content in keratin, and therefore, their respective disulfide bonds, it is believed that other enzymes, such as disulfide reductases, are likely to be involved in this process, acting in addition to keratinolytic proteases [28]. Although there are many publications describing keratinase characterization, the molecular mechanism of microbial keratinolysis is not yet completely elucidated.

Genomic analysis of keratinolytic bacteria is seldom reported [29,30,31,32,33,34]. *Streptomyces* species are one of the main producers of keratinases [35], but so far, there is only one recently reported genome of keratinolytic-degrading *Streptomyces* [36]. Moreover, according to our knowledge, there are no comparative studies between keratinolytic and non-keratinolytic bacteria, which may unveil the underlying genetic factors contributing to its degradative capacity. Recently, our Chilean marine actinobacterial culture collection [37,38,39] was screened for extracellular enzyme activities analyzing 75 strains [40], providing evidence of promising keratinolytic activity. To see whether we could bioinformatically predict the genes encoding putative keratinases, in this study, we sequenced the genome of three of these streptomycetes, selected for showing varying levels of keratin degradation: Strains G11C, CHD11, and Vc74B-19, with high, low, and no keratinolytic activity, respectively. We present an integrative comparative analysis between the universe of putative proteases belonging to these strains, complementing information from orthogroups of proteases, peptidase families, cell location prediction, and phylogeny, to finally deliver a bioinformatic prediction consisting of a set of genes considered to encode potential keratinases. Efforts addressing a thorough *in silico* investigation of putative proteases and peptidases, to obtain those likely to be involved in feather degradation, have been accomplished for the first time. Interestingly, 18 of 24 proteases predicted by this pipeline (6 of seven potential keratinases (predicted by phylogenetic analysis), and 12 of 17 unique genes of the G11C genome (“unassigned p-orthogroup” peptidases)) and thought to be involved in keratinolytic activity were validated in our previous study by being present in the secretome of strain G11C [40]. This study provides a not so far described bioinformatic multi-step pipeline, that helps decipher potential genetic factors that enable some strains to have higher efficiency in keratin degradation. 

## 2. Results

### 2.1. Genome Features of Streptomyces Strains with Differential Keratinolytic Activity

Three strains (Appendix A) from our Chilean marine actinobacterial culture collection [37,38,39] were selected for genome sequencing, based on a previous feather degradation screening [40]: *Streptomyces* sp. G11C, presented the highest level of keratinolytic activity, exhibiting complete feather degradation in liquid culture; *Streptomyces* sp. Vc74B-19, evidenced no keratinolytic activity, leaving the feather structure unaltered, and *Streptomyces* sp. CHD11 presented a low level of degradation, showing the partial dissolution of the fibers of the feathers [40]. To obtain insights into the genetic determinants involved in feather degradation, genomes were sequenced using the Illumina NextSeq platform with a 150 bp × 2 configuration. Final assembly statistics, including quality metrics, for each genome, are displayed in Table 1. CHD11 and Vc74B-19 genomes, of approximately 7.5 Mbp, are larger in size than the G11C genome (nearly 6.9 Mbp), although the GC content is similar for all three strains and is comparable with percentages found in actinobacterial strains [41]. 

Previous taxonomic classification using the 16S rRNA gene indicated that these marine strains belong to the *Streptomyces* genus [37]. To obtain a precise taxonomic placement, we performed a phylogenomic analysis (Figure 1, Appendix A) and calculated average nucleotide identity (ANI) (Appendix A). In the case of the phylogenomic tree, the inference was made with Orthofinder [42] using 634 single-copy orthogroups. These analyses indicate that both streptomycete strains presenting either low and no keratinolytic activity, strains CHD11 and Vc74B-19, respectively, are phylogenetically close, grouping in the same sub-clade with the closest neighbor *Streptomyces emeiensis* CGMCC 4.3504^T^ [43]. This is consistent with ANI values obtained between strains Vc74B-19 and CHD11 (88.2%) and with *S. emeiensis* type strain—87.5% for strain CHD11 and 87.4% for strain Vc74B-19. On the other hand, strain G11C, with high keratinolytic activity, groups with *Streptomyces albidoflavus* NRRL B-1271^T^ [44], consistent with an ANIm of 96.1% that indicates strain G11C could be considered as part of the *albidoflavus* species [45]. A strain belonging to the *albidoflavus* species has been described in the literature to possess keratinolytic activity, secreting at least six extracellular proteases when cultured on a feather meal-based medium [14]. Altogether, these findings motivated us to study the minor differences between our genomes and examine a possible bioinformatic explanation for the observed differential keratinolytic activities.

### 2.2. Comparative Genomics of Differential Keratinolytic Streptomycete Strains

To identify potential genes encoding keratinases that could be involved in keratin hydrolysis and the observed differences in *Streptomyces* strains G11C, CHD11, and Vc74B-19, a comparison of the diversity and quantity of peptidases present in all three strains was accomplished using the following approaches: Genome annotation using the Prokka, PANNZER2, and eggNOG servers, classification by MEROPS database and identification of orthologous groups. In addition, to compare these protein sequences with functionally characterized keratinases obtained from literature, a similarity network was developed.

#### 2.2.1. Protease Search

Multiplatform annotation revealed that all three *Streptomyces* strains presented a similar abundance of genes encoding for putative peptidases, corresponding to 3% of their genomic content (Table 2, Appendix A). According to MEROPS classification, approximately 85% of their putative peptidases were classified into a protease family (Appendix A), where strains G11C, CHD11, and Vc74B-19 presented 46, 48, and 49 peptidase families, respectively. In general, the most substantial fraction of peptidases for all three strains belongs to the serine (47–48.6%) and metallo- super-families (36.7–41.1%), whereas the minor fraction consists of cysteine (6.6–11.4%), aspartic (1.1–1.3%), threonine (1.1–1.3%) and mixed (1.1–1.3%) super-families (Appendix A). This result is consistent with previous studies, where serine, metallo-, and cysteine peptidases are the dominant proteolytic enzymes (>90%) of Bacteria, and aspartic and threonine peptidases contribute to a minor extent [46,47]. Interestingly, *Streptomyces* sp. G11C presents two putative peptidases belonging to two unique families (C02 and S53) that are not present in strains CHD11 and Vc74B-19. C02 belongs to the cysteine peptidases of the calpain family, although its biological role in bacteria is unclear, and there are not many studies that clarify its properties [48]. On the other hand, serine S53 family peptidases belong to the sedolisin family, which has been strongly correlated with an acidophilic lifestyle [46]. In contrast, strains CHD11 and Vc74B-19 share five peptidase families (C14, C15, C40, C56, and M103) between them that are absent in strain G11C. Similarly, strain G11C only shares a peptidase family (M17) with the non-keratinolytic strain Vc74B-19, which is not present in strain CHD11. The similarities shared with the non-keratinolytic strain Vc74B-19 could indicate that such peptidases may not contribute to the degradative ability of the keratinolytic strains G11C and CHD11.

Subsequently, putative peptidases (584 sequences) of the three strains were classified into 140 protease orthologous groups (p-orthogroups) using Orthofinder [42]. As expected, putative peptidases from the same p-orthogroup belong to a single peptidase family (Appendix A). In fact, 102 p-orthogroups are shared by all three strains (Figure 2), confirming the similarity of the protease space between these streptomycete genomes. As for strain G11C, three p-orthogroups are exclusively shared with strain CHD11, and eight p-orthogroups with Vc74B-19, that are not present in the other strain. Particularly, the three p-orthogroups 121, 122, and 134 (Appendix A), shared between the keratinolytic strain G11C and the low-keratinolytic strain CHD11, belonging to the M50, S12, and S01 peptidase families, can be considered good candidates for potential keratinolytic proteases, assuming both strains could have similar degradation mechanisms. By contrast, the eight p-orthogroups shared between the keratinolytic strain G11C and the non-keratinolytic strain Vc74B-19 can be potentially discarded as keratinolytic peptidase candidates. 

Most of the shared p-orthogroups belong to the serine (n = 37) and metallo- (n = 40) super-families, while the cysteine, aspartic, threonine, and mixed super-families are found in a smaller proportion (n = 2–5), which agrees with our previous MEROPS results. On the other hand, some peptidases are not classified into any p-orthogroup. Further exploration of these “unassigned p-orthogroup” peptidases could give an insight into the differences observed in terms of keratinolytic activity between the strains, considering that strain G11C presents comparatively the greatest keratin degradative capacity under the conditions analyzed [40]. Among the 17 putative peptidases unique for strain G11C, there are five serine peptidases (families S01, S16, S15, S53, and S51), three metallo-peptidases (families M86, M50, and M56), two cysteine peptidases (families C82 and C02), and seven unassigned peptidases to any family.

Putative proteases belonging to families, S01, S08, and M04, where most known bacterial keratinases are found (Appendix A), can serve as indicators to search for putative keratinolytic proteases. Two sequences of *Streptomyces* sp. G11C belonging to the S01 family draw our attention, which are absent in orthogroups shared with the non-keratinolytic strain Vc74B-19: An unassigned p-orthogroup peptidase (G11C_00267) and a peptidase (G11C_00756) belonging to one of the three p-orthogroups shared between the keratin degrading strains G11C and CHD11. Additionally, *Streptomyces* sp. G11C presents 8, 12, and 3 putative peptidases belonging to the families S08, S01, and M04, respectively, that may be of interest for the search for putative keratinases, despite belonging to orthogroups shared between the three strains. The detail of the promising sequences of *Streptomyces* sp. G11C (i.e., “unassigned p-orthogroup” peptidases, peptidases shared between the strains G11C and CHD11, and peptidases belonging to the peptidase families S01, S08, and M04) are summarized in Appendix A.

#### 2.2.2. Network Analysis

To complement the previous analysis, and identify those sequences related to families of known keratinases, a similarity network was constructed using an all-vs-all local alignment. For this analysis, 584 putative proteases of the three strains (hereinafter named “three-strain dataset”), 61 functional keratinases (mainly from Gram-positive bacteria) collected from NCBI (Appendix A), and 50 selected trypsin, papain, and pepsin sequences, representing our hypothetical non-keratinase database (Appendix A), were compared. Nodes in the network depict each protease, and an edge represents a hit in the resulting alignment (Figure 3; Appendix A). Our protease similarity network graphically depicts the p-orthogroup distribution and identifies p-orthogroups that are related to functionally described keratinases. In total, 123 network communities composed of at least two or more nodes were detected by the Louvain algorithm.

Three network communities (N° 1, 4, and 41) possess sequences linked with functional keratinases with an E-value threshold of 1 × 10^−40^ (Appendix A). The largest one, community N° 1 (Figure 3B), is related to sequences belonging to the peptidase family S08, constituted by most of the functional keratinases (n = 51 sequences) harboring 3, 5, and 6 sequences from strains G11C, CHD11, and Vc74B-19, respectively (p-orthogroups 0 and 11). Community N° 4 (Figure 3C) is composed of putative peptidases belonging to the family S01, where strains G11C, CHD11, and Vc74B-19 contribute with 5, 6, and 6 sequences, respectively (p-orthogroups 8, 17, and 116). This cluster presents six functional keratinases from *Actinomadura*, *Streptomyces*, and *Nocardiopsis*. Curiously, two putative non-keratinase sequences, annotated as trypsin-like serine protease (family S01) from other *Streptomyces* strains, also cluster together. This observation suggests that these two specific peptidases could have keratinolytic activity, although it has not been experimentally tested yet. Finally, in the smaller community N° 41 (Figure 3D), sequences belonging to the peptidase family M04 can be observed, consisting of 3, 2, and 5 sequences from strains G11C, CHD11, and Vc74B-19, respectively (p-orthogroups 16 and 64), clustering together with a functional keratinase from a *Geobacillus* strain (AJD77429.1). Scattered in the network, we found three keratinases from *Lactobacillus* and *Bifidobacterium* that do not group with any sequence from our three-strain dataset. In general, putative non-keratinase sequences are depicted as unconnected nodes or communities, except for specific cases such as communities N° 4 (mentioned above) and N° 47, both belonging to the peptidase family S01 (Appendix A).

In summary, there are 11, 13, and 16 putative proteases linked to functional keratinases from *Streptomyces* strains G11C, CHD11, and Vc74B-19, respectively. Although the number of genes is less for keratinolytic-strain G11C, the percentage of identity when compared with some functional keratinase sequences is higher with this strain (Appendix A). For instance, the first hit for all three strains is a keratinase synthesized by *Streptomyces albidoflavus* TBG-S13A5 (AYM48028.1), which presented a 96.4% amino acid identity with a predicted protein from strain G11C (G11C_01512), 72.8% identity for strain CHD11 (CHD11_02603) and 72.2% for strain Vc74B-19 (Vc74B-19_03690). The sequence found in G11C, belonging to peptidase family S01, p-orthogroup 8, and community N° 4, can be considered a putative keratinase, considering the high amino acid similarity with the known keratinase from *Streptomyces albidoflavus* TBG-S13A5, which can also be explained by the phylogenomic closeness between both strains. 

### 2.3. Filtering by Cellular Localization Scores

Cellular localization data and phylogeny were employed to narrow the list of candidate keratinases that could potentially make a difference in keratin degradation for strain G11C. Proteases secreted into the extracellular medium are expected to play an important role in keratin degradation [2]. Thus, to identify those sequences predicted as extracellular, all previously mentioned datasets (three-strain set, functional keratinases, and putative non-keratinases) were inputted into three localization prediction software: PSORTb, CELLO, and SignalP (Appendix A). To add informative categories and therefore, facilitate interpretation of results in the following steps, we separated the three-strain set into: (i) Keratinase-linked proteases (40 sequences), which are sequences linked to functional keratinases according to the network, (ii) “unassigned p-orthogroup” sequences (28 sequences) according to protease orthogroup classification, and (iii) the remaining into the three-strain category (516 sequences). Principal component analysis (PCA) and t-distributed stochastic neighbor embedding (t-SNE) were employed to embed sequences into a bidimensional space through the numerical scores retrieved from each tool (Figure 4). Two semi-defined clusters are visualized in the PCA plot (Figure 4A), representing intracellular (lower left corner) and extracellular (lower right corner) proteases. As expected, most of the keratinase-linked proteases from the three-strain dataset group together with known keratinases are represented as extracellular proteases. On the other hand, most of “unassigned p-orthogroup” sequences, including the peptidases unique of *Streptomyces* sp. G11C, are predicted as cytoplasmic and membrane-associated proteases. Non-keratinases were distributed in sparse coordinates, where seven sequences can be considered putative extracellular proteases.

Sequences were further clustered in the t-SNE bidimensional space using the DBSCAN algorithm [49], revealing a clear separation of the sequences into groups (Figure 4B, Appendix A). This clustering also guides the comparison of the sequence space representation between the PCA and the t-SNE. From Figure 4B, three DBSCAN clusters related to extracellular localization characteristics can be visualized. These were named t-SNE group 0, 1, and 2, each one with 68, 91, and 32 total sequences, respectively, where a considerable number of functional keratinases is present (Table 3; Appendix A). Therefore, these groups are of great interest as they may contain promising sequences to encode potential extracellular keratinases. In all the identified groups, there are more sequences predicted as extracellular belonging to strains Vc74B-19, and CHD11 than from strain G11C. Possibly, differences in keratinolytic activity could be due to specific characteristics at the amino acid sequence level. To deepen our analysis, we complemented this study with a phylogenetic analysis of the sequences identified in the t-SNE groups.

### 2.4. Phylogeny on t-SNE Groups

To identify extracellular peptidases of *Streptomyces* sp. G11C that are phylogenetically related to functional keratinases, multiple sequence alignments (MSA), and phylogenetic analysis were performed on the t-SNE groups 0, 1, and 2 (containing sequences predicted as extracellular). The average occupancy (average number of residues per position in the alignment) [50] of the three t-SNE groups MSAs was rather low as highly divergent sequences hinder phylogenetic tree construction: group 0—35.8%, group 1—19.3%, and group 2—35.7%. For this reason, we filtered positions with occupancy below 70% to generate more compact MSAs, and the new average occupancy for these compact alignments was t-SNE group 0—87.4%, group 1—85.1%, and group 2—90.3% (Appendix A). Then, we constructed bootstrapped maximum likelihood trees with these compact MSAs (Figure 5). Due to the high divergence observed for the sequences in each t-SNE group, an additional tool was applied, ancestral state reconstruction, to aid the interpretation of the trees.

The same discrete categories used in the PCA and t-SNE plots (i.e., functional keratinase, keratinase-linked protein, three-strain category, and non-keratinase) were used for the ancestral state reconstruction analysis. With this tool, a probability distribution is assigned to each ancestor node within the tree, indicating the likeliness of the ancestor to belong to one of these categories. This provides a visual interface to study which branches could potentially be related to proteases with keratinolytic activity, as the ancestral state depends on both branch length (i.e., sequence similarity) and tree topology [51,52]. We applied three filters to select clades for a more detailed description. First, we analyzed branches that presented >50% probability of belonging to the functional keratinase or keratinase-linked categories. Second, we selected clades that possess at least one functional keratinase. And third, we focused only on clades where at least one sequence from strain G11C is present. These criteria reduced the subsequent analysis to clade 2 in the t-SNE group 0 (Figure 5A, Box 2) and clades 1 and 5b from t-SNE group 1 (Figure 5A, Box 1 and 5b). Clades of the t-SNE group 2 did not meet these requirements, and therefore, were not analyzed (Appendix A). 

Clade 2 of t-SNE group 0 (Figure 5A, Box 2) harbors seven proteases from the three-strain set, annotated as serine proteases belonging to the S01 family, community 4, three from the p-orthogroup 17, and four from the p-orthogroup 8. Within this clade, there are four keratinases, belonging to *Actinomadura* (ASU91959.1, AMH86070.1), *Nocardiopsis* (AAO06113.1), and *Streptomyces* (CAH05008.1) strains. Interestingly, a sequence of the keratinolytic strain G11C (G11C_05333) and low-keratinolytic strain CHD11 (CHD11_00976) are phylogenetically close, compared to the non-keratinolytic strain Vc74B-19. It is possible that specific motifs or amino acids within these sequences, not present in strain Vc74B-19, enhance the activity of the codified enzymes, and should be considered as candidates for putative keratinases. In addition, a subclade that presents Vc74B-19 and CHD11 sequences related to the known *Streptomyces fradiae* K11 keratinase CAH05008.1 was observed. In this case, branch lengths indicate significant sequence divergence, and therefore, no evidence of putative keratinolytic activity can be assigned to the three-strain genes of this subclade.

In the case of the t-SNE group 1 tree, clade 1 (Figure 5B, Box 1) groups seven three-strain proteases belonging to the p-orthogroup 0, community 1: Three from strain G11C (G11C_02264, G11C_03013, G11C_05273), two from strain CHD11 (CHD11_02120, CHD11_02299), and two from strain Vc74B-19 (Vc74B-19_00125, Vc74B-19_05629). All these sequences are annotated as serine proteases of the S08 family. There are only two sequences of functional keratinases within this clade, one is a partial sequence from *Streptomyces* sp. OWU 1633 (AAU94350.1) and the other is from *Amycolatopsis* sp. BJA-103 (QGA70043.1), both belonging to the family S08. In this clade, the sequences of CHD11 and Vc74B-19 strains are phylogenetically closer, compared to strain G11C. 

Clade 5b (Figure 5B, Box 5b) comprises five three-strain sequences of the p-orthogroup 8, community 4: Three from strain G11C (G11C_01510, G11C_01512, G11C_02546), one from strain CHD11 (CHD11_02602) and one from strain Vc74B-19 (Vc74B-19_03689), annotated as streptogrisins A, B, and D. This clade has a branch that contains two keratinases from *Streptomyces albidoflavus*, strains Fea-10 (AQX39246.1) and TBG-S13A5 (AYM48028.1). In this branch, only one sequence of strain G11C is present (G11C_01512), and given its high similarity with the mentioned keratinases (96.4% amino acid identity), it is probably a potential keratinase, which could explain, to a certain extent, the differences in keratinolytic activity between our strains. All these sequences belong to the peptidase S01 family. 

Focusing on *Streptomyces* sp. G11C, which evidenced the greater level of keratin degradation, we identified seven putative proteases of interest, that are phylogenetically close to known keratinases, and that could contribute to explaining the observed differential keratinolytic activities. These sequences are the following: G11C_05333, G11C_02264, G11C_03013, G11C_05273, G11C_01510, G11C_01512, and G11C_02546. They belong to p-orthogroups 0, 8, and 17, communities 1 and 4, which are related to peptidase families S01 and S08, therefore, supporting this prediction.

### 2.5. Summary of the Main Results of the Bioinformatic Pipeline

To determine the genetic determinants potentially involved in the keratinolytic capacity of marine *Streptomyces*, a comparative genomic analysis was carried out between three streptomycetes with different keratinolytic activities, strains G11C, CHD11, and Vc74B-19 with high, medium, and null activity, respectively (Figure 6). Initially, a search for proteases was performed by genomic annotation using three servers: Prokka, PANNZER2, and eggNOG. These proteases were manually cured using Blastp and classified into peptidase families according to the MEROPS database. Subsequently, a comparative genomic analysis between the universe of proteases of the three strains allowed the identification of protease orthogroups shared between the three streptomycete genomes, highlighting proteases unique to each strain. From this analysis, genes of interest were exclusively related to keratinolytic strains, where 17 genes unique to strain G11C and three peptidases belonging to orthogroups shared between strains CHD11 and G11C are found. Additionally, a similarity network analysis of all the proteases of the three strains together with two databases of functional keratinases and hypothetical non-keratinases, allowed the identification of three communities related to functional keratinases belonging to peptidase families S01, S08, and M04. 11 proteases from *Streptomyces* sp. G11C emerge from this analysis. Subsequently, to identify extracellular proteases related to keratinases, the information from the similarity networks and p-orthogroup analysis was integrated with a cell localization analysis through t-SNE based clustering. This analysis allowed the identification of three groups containing extracellular proteases, named groups t-SNE 0, 1, and 2. In this analysis, most of the unique peptidases of the strain G11C (unassigned p-orthogroup peptidases) and the peptidases shared between the keratinolytic strains CHD11 and G11C were predicted to be intracellular. However, given their exclusive relationship with the keratinolytic strains, they are still considered interesting because their presence may possibly explain the differences in keratinolytic activity between the three strains. On the other hand, the peptidases predicted to be extracellular present in the t-SNE groups were subjected to a phylogenetic analysis incorporating the tool: Ancestral state reconstruction, which assigns a probability distribution to each ancestor node within the tree, of belonging to the categories: Functional keratinase, keratinase-linked protein, three-strain category, and non-keratinase. Finally, after applying selection criteria for the analysis of the clades of the phylogenetic trees (presence of a functional keratinase, presence of a G11C sequence, and 50% probability in the ancestor node of being keratinase or keratinase-linked sequence), seven coding sequences for potential extracellular keratinases were identified in *Streptomyces* sp. G11C. 

In this analysis, sequences codifying for (1) proteases related exclusively to keratinolytic strains and (2) proteases predicted to be extracellular and related to functional keratinases are considered as interesting candidates that could explain the differences in keratinolytic activity between the three strains.

## 3. Discussion

In this study, a multi-step bioinformatic pipeline, applying several comparative genomics tools, was developed to predict a set of genes that encode putative keratinases in a marine *Streptomyces* strain with keratinolytic activity. To see if genetic features related to these activities could be identified, three strains with differential keratinolytic activity were selected, *Streptomyces* sp. G11C presented a rather high percentage of feather degradation, reaching approximately 80% and with a relative keratinase activity of 60%, after five days of incubation, whereas *Streptomyces* sp. CHD11 presented lower relative keratinase activity (less than 10%) [40]. In contrast, *Streptomyces* sp. Vc74B-19 presented no keratinolytic activity, even after 10 days of incubation [40]. 

Genome comparison showed approximately 3% of the total gene count encoded putative peptidases, revealing an unexpected similar abundance and diversity in all three strains. Previous reports described that among bacterial species, the percentage of peptidases encoded in the genomes ranges from 1.5% to 4% [53]. A similar diversity of peptidase families was found in all three strains, with serine, metallo- and cysteine super-families being more abundant in all three genomes. Peptidase diversity may reflect their adaptation to environmental conditions [46]. Considering the marine origin of the *Streptomyces* analyzed, the diversity of peptidases could be related to environmental characteristics such as the pH of the ocean, which varies from slightly neutral to alkaline [54]. Serine, cysteine, and metallo-peptidases are generally active under these conditions [55,56] and may contribute to their ecologic success, and possibly, their degradative abilities. In general, strains CHD11 and Vc74B-19 showed to be more similar to each other, sharing a large number of peptidase families and p-orthogroups that are absent in strain G11C, in agreement with their close phylogenomic relatedness. By contrast, strain G11C presented more “unassigned p-orthogroup” peptidases (17 unique peptidases) compared to the other two strains CHD11 and Vc74B-19 (8 and 3, respectively), of which 12 were actively involved in keratin degradation, whose presence was confirmed in our previous secretome analysis in keratinolytic strain G11C [40], highlighting them as interesting candidates. According to the subcellular localization analysis, most of these sequences are predicted as cytoplasmic and membrane-associated proteases. Although most of the keratinolytic proteases known to date are predominantly extracellular, some cell-bound and intracellular keratinolytic proteases have also been described [57]. Therefore, the participation of the unique proteases of the strain G11C in the keratinolytic activity cannot be ruled out.

Most of the keratinases known to date, have been classified as serine proteases [14,16,19,58,59], and a few as metalloproteases [20,60,61]. The latter mainly come from Gram-negative bacteria and fungi [62]. To compare and identify promising sequences related to known keratinases, we constructed a functional keratinase database, mainly from Gram-positive bacteria (Appendix A), retrieving data from the NCBI. An analysis based on the MEROPS classification indicates that keratinases from *Bacillus* belong to the peptidase S08 family, and keratinases from *Actinobacteria*, such as *Nocardiopsis*, *Actinomadura*, and *Streptomyces*, belong mostly to the S01 family. There is scarce information about the metalloprotease family associated with keratinases. In our database, only one of the keratinases, from a *Geobacillus* strain (AJD77429.1), belongs to the metalloprotease super-family, specifically to the family M04. A protease similarity network, using these functionally characterized keratinases together with our three streptomycete genomes and non-keratinases databases, highlighted three communities of nodes containing keratinase-linked peptidases belonging to the S1, S8 (serine proteases), and M4 (metalloprotease) families, allowing the identification of 11 promising sequences of the keratinolytic strain G11C. It is worth mentioning that community 41 (family M04) presented the least number of grouped sequences, which is possibly related to the presence of only one functional keratinase, as mentioned above. Furthermore, the three strains show few peptidases belonging to this family, for example, *Streptomyces* sp. G11C only presents three M04 metallopeptidases that were subsequently discarded because they did not meet the pipeline criteria (being extracellular and phylogenetically close to functional keratinases). On the other hand, the promising sequences, including the unique peptidases of the strain G11C (unassigned p-orthogroup peptidases), and the peptidases belonging to p-orthogroups shared between the keratinolytic strains CHD11 and G11C, did not group in any community of the network, since most belong to other protease families, being divergent from the known keratinase families. In these groups, there are only two serine proteases of the family S01, although they did not group in community 4 (family S01). In addition, it can be mentioned that one of these unique sequences of strain G11C (G11C_00267) presented high abundance in the secretome analysis carried out previously [40], indicating a relevant role in the degradation of keratin. 

To narrow the search for potential keratinases, we complemented this information with the subcellular localization data and phylogeny. In the t-SNE clustering analysis, we identified those putative peptidases that are predicted as extracellular (including the keratinase-linked sequences mentioned above). In this analysis, we reasoned that cellular localization prediction is intimately related to a putative keratinolytic function. This is in line with previous reports, that have shown that the macromolecular characteristics of keratin prevent its direct absorption by microbial cells [2]. Thus, the utilization of keratin as a nutrient source usually requires the production of extracellular keratinases. Therefore, to finely separate and select potential extracellular keratinase sequences, we included a phylogenetic analysis of the clusters, which also provides a way to integrate the similarity network data into our selection process. The latter is one of the main challenges addressed in this work, since network sequence data has been mainly used as a standalone tool for functional inference [63], partly due to its graphics that are not completely compatible with tabular or structural sequence data. Finally, from this analysis, we identified seven gene sequences encoding potential keratinases (belonging to families S01 and S08) together with 17 unique genes encoding unassigned p-orthogroup peptidases in *Streptomyces* sp. G11C could explain the differences observed, in terms of keratinolytic activity, between the three strains. Apparently, the degradation of recalcitrant keratin wastes requires the cooperation of several keratinolytic proteases, as evidenced in some bacterial [36,64,65,66,67] and fungi species [7,68]. Recently, a work reported by Huang et al. [65], evidenced the presence of five proteases in the culture of *Bacillus* sp. 8A6 with keratin-rich substrates, belonging to four protease families M12, S01A, S8A, and T3. In fungi, the participation of a set of proteases in keratin degradation has also been reported. Pathogenic fungi mainly secrete endoproteases, including proteases from families A1, S8A, M36, and M35 [68]. The non-pathogenic fungus *Onygena corvina* secretes proteases belonging to three protease families: S08, M28, and M03, when cultivated with pig bristles [7]. However, we have not found proteases belonging to these families in our analysis with streptomycete genomes, except for some putative peptidases belonging to the M28 family, suggesting that the mechanisms of keratin degradation vary between fungi and bacteria. For most keratinolytic studies involving *Streptomyces*, the approach has been focused on purifying and characterizing the main keratinolytic enzyme. For example, in work reported by Bressollier et al. [14], at least six extracellular proteases were identified in the culture of *Streptomyces albidoflavus* grown on feather meal-based medium, but only the most abundant keratinolytic serine protease was further characterized. Recently, through a transcriptomic analysis performed by Li et al. in 2020, it was possible to elucidate a set of factors involved in the keratin degradation mechanism mediated by *Streptomyces* sp. SCUT-3 [36]. In this analysis, 19 genes codifying potential extracellular proteases, along with 10 genes codifying potential intracellular proteases belonging to serine-type, cysteine-type, and metalloproteases, were up-regulated during growth in medium containing feathers. In addition, two genes involved in mycothiol synthesis, and some genes related to sulfite production were also up-regulated, indicating a cooperative action of reducing agents in the breaking of feather disulfide bonds. According to this evidence, it is conceivable to propose a set of enzymes acting together in keratin degradation by a single bacterium. The literature mentioned above is solely based on the functional exploration of keratin degradation, and no systematic bioinformatics analysis has been approached. The advantage of our pipeline is it considers whole genomes instead of single sequences, which has not been previously addressed with keratinases. With the advances in genome sequencing and the improvement of the number of genome-scale studies, our pipeline could be enriched and further bioinformatic predictions tested. 

To go further on this argument, in our recent work [40], we confirmed the presence of these predicted enzymes in the *Streptomyces* sp. G11C secretome: Six of seven keratinolytic proteases (predicted by phylogenetic analysis), and 12 of 17 unique genes of the G11C genome (“unassigned p-orthogroup” peptidases), were detected under the culture conditions with feathers as the sole carbon source, indicating that a set of enzymes may act synergistically during keratin degradation. Interestingly, one of the unique genes of the G11C strain (G11C_00267) presented one of the highest protein abundances in the proteomic analysis [40], suggesting an important role for this enzyme in the keratin degradation mechanism. These findings are consistent with our bioinformatic predictions. The coordination of all these extracellular and intracellular enzymes, including the unique peptidases of *Streptomyces* sp. G11C could be potentiating its keratinolytic activity, leading to an advantage over the other *Streptomyces* analyzed strains CHD11 and Vc74B-19. Genes encoding disulfide reductases and genes related to sulfite export were similar in the three strains (data not shown), suggesting that disulfide bond reduction is not related to the observed functional differences, at least not the common mechanisms described in the literature [35]. Additional efforts to discover the reasons for such functional differences between these strains are part of our ongoing investigation. Our novel bioinformatic pipeline, together with the increased sequencing of keratinolytic strains genomes, could serve as the basis for future predictions of keratinolytic proteases, facilitating the selection of potential keratinolytic bacteria. To our knowledge, this is the first comprehensive bioinformatics analysis that complements comparative genomics with phylogeny, network similarities, and cellular localization prediction to provide a set of candidate genes considered to encode putative keratinases.

## 4. Materials and Methods

### 4.1. Bacterial Strains

Previously, bacterial strains belonging to our Chilean marine actinobacterial culture collection, isolated from marine sediments, sponges, and sea urchins collected from the coast of Chile [37,38,39], were analyzed for keratinolytic activity through a simple feather degradation test on agar plates and culture tubes [40]. Based on these results, three *Streptomyces* were chosen to perform a genomic comparison according to their keratinolytic activity. *Streptomyces* sp. G11C isolated from marine sediments derived from Penas Gulf, with high keratinolytic activity, *Streptomyces* sp. CHD11, isolated from a marine sponge from Chañaral de Aceituno Island, with low keratinolytic activity and *Streptomyces* sp. Vc74B-19, isolated from marine sediments from Valparaíso Bay, with no keratinolytic activity (Appendix A). 

### 4.2. Genomic DNA Extraction

Strains were grown on ISP2 media for 24–48 h, and cells were collected by centrifugation (13,000× *g*, 2 min), resuspended in 50 mM EDTA, and then mechanically separated with a shank. For cell rupture, disaggregated cells were treated with Lysozyme 10 mg/mL, Lysostaphin 10 mg/mL, and Proteinase K 1 mg/mL for 1 h at 37 °C. Then, for DNA purification, the Promega Wizard DNA extraction kit was used, according to the manufacturer’s description. Finally, an alcohol treatment was used for DNA precipitation and cleaning, with 100% isopropanol for DNA precipitation, followed by 70% ethanol to wash the DNA pellet. The sample was rehydrated and stored according to the Promega Wizard DNA Extraction Kit instructions.

### 4.3. Genome Sequencing, Assembly, and Annotation

Sequencing was performed using the Illumina NextSeq platform with a 150 bp × 2 configuration. Raw reads were examined for coverage and trimmed for a minimum quality of 20 using Sickle v.1.33 [69]. Genome assembly was performed using SPAdes v.3.13.1 with default parameters [70,71] and Unicycler in normal mode [72] (internally using SPAdes v.3.11.1. The SPAdes assembly pipeline contains a final step of scaffolding using SSPACE [73]. Assembly quality was measured using QUAST v.5.0.2 [74], and genome completeness and contamination were evaluated using CheckM v1.0.18 [75]. A unique working assembly for each strain was selected based on the one with a smaller number of contigs, largest contigs (N50, L50 evaluation), and the lowest contamination metrics. Assembled genomes were annotated using Prokka v.1.13.4 [76], PANNZER2 [77], and eggNOG 5.0 [78] web servers. 

### 4.4. Phylogenomic Tree Inference

*Streptomyces* genomes were selected from the tree described in Nouioui et al. [79], spanning over subclades. The selection criteria used was genome quality (fewer contigs and higher completeness based on CheckM), and type strains were preferred for the analysis. Assembly metrics were obtained using QUAST v.5.0.2 and CheckM v.1.0.18. The genomes of strains that closely resembled our strains by 16S rRNA identity [37] were also added: *Streptomyces albidoflavus* NBRC 13010 (=NRRL B-1271) and *Streptomyces aurantiogriseus* NRRL B-5416. *S. albogriseoleus* genome was not included, since assemblies presented anomalies, according to NCBI Genbank. The genomes of a *Catenulispora* strain (DSM 44928) and a *Kitasatospora* strain (KM-6054) were used as outgroups. Selected genomes and the outgroup genomes were retrieved from the Pathosystems Resource Integration Center (PATRIC) database and are described in Appendix A. Phylogenomic reconstruction was performed using Orthofinder v.2.3.11 [42], with FastTree v.2.1.10 [80], in multiple sequence alignment mode (-M msa option), employing the default parameters for the orthogroup definition. Complementary to the phylogenomic analysis, average nucleotide identity calculation was performed using all of the selected genomes (Appendix A), with the Python PyANI package [81] and depicted using the seaborn package in Python.

### 4.5. Putative Protease Identification

Annotations obtained from Prokka, PANNZER2, and eggNOG were consolidated in Appendix A. Putative proteases were identified by matching annotations with the following keywords: Peptidase, protease, proteinase, sortase, caspase, penicillin-binding protein, insulinase, snapalysin, and mycosin. Each coding sequence that presented a keyword in at least one of the predictions was manually checked using BLASTp v.2. against the NR database, to confirm the proteolytic nature of the enzyme. In this work, the terms “protease” and “peptidase” have been used indistinctly [82].

### 4.6. Classification of Protease Families and Identification of Protease Orthogroups

For the classification into protease families, a reciprocal best hit search (E-value threshold, 1 × 10^−20^ of the three-strain dataset was performed against MEROPS, the peptidase database, release 11, using the HMMER-based web-server utility [83]. This database classifies peptidases into seven superfamilies based on the catalytic residue (Aspartic (A), Cysteine (C), Glutamic (G), Metallo (M), Asparagine (N), Serine (S), Threonine (T)) along with two superfamilies of Mixed (P) and Unknown (U) catalytic types, and further divides these superfamilies into 255 proteolytic families based on similarities in amino acid sequences [84]. Additionally, the curated list of proteases for the three strains was grouped into orthogroups using Orthofinder [42] with the default E-value 1 × 10^−40^ These were named protease orthogroup or p-orthogroup. Venn diagram visualization of common p-orthogroups between strains was done using the package eulerr in R [85].

### 4.7. Creation of Custom Databases: Functional Keratinases and Putative Non-Keratinases

To identify putative keratinases in the genomes, two custom databases with sequences of functional keratinases and putative non-keratinases were constructed by retrieving sequences from the NCBI NR database (query date: March 2020). The functional keratinase database was built according to literature and contains 61 sequences from *Actinomadura*, *Amycolatopsis*, *Bacillus*, *Bifidobacterium*, *Brevibacillus*, *Deinococcus*, *Geobacillus*, *Lactobacillus*, *Meiothermus*, *Nocardiopsis*, *Streptomyces*, and *Thermus* (Appendix A). In addition, to differentiate and discard those sequences that do not encode possible keratinolytic proteases, a putative non-keratinase database was created. Several reports indicate that keratin has high stability against common proteases, such as pepsin, papain, and trypsin [1,2]. Based on this information, a hypothetical non-keratinase database was constructed (Appendix A). This database contains 50 protein sequences belonging either to the trypsin, papain, or pepsin families. All sequences are derived from Gram-positive bacteria belonging to the genera *Bacillus*, *Bifidobacterium*, *Brevibacterium*, *Chloroflexi*, *Clostridium*, *Corynebacterium*, *Enterococcus*, *Lactobacillus*, *Listeria*, *Staphylococcus*, *Streptococcus*, and *Streptomyces*.

### 4.8. Similarity Network

For the similarity network of the three-strain dataset, a blastp all-vs-all output was processed in Python using the networkX package [86]. Both functional keratinase and non-keratinase databases were added to the three-strain dataset for network construction. In the network, nodes correspond to each protease and edges to a hit in the resulting alignment. The E-value cutoff for the display of the edge weights was set to 1 × 10^−5^ 1 × 10^−20^, 1 × 10^−40^, and 1 × 10^−80^, being 1 × 10^−40^ the value chosen for the analysis, since it is a typical value (order of magnitude) employed in sequence similarity networks [87], and subsequent steps in the pipeline have small sensitivity to using stricter cutoffs (Appendix A). Community detection was performed using a weighted Louvain algorithm with a default resolution parameter of 1 [88]. A depiction of the network was obtained using Gephi v.0.9.2 [89] with a combination of Fruchtermann-Reingold [90] and Yifan Hu [91] layout algorithms. 

### 4.9. Cellular Localization and Dimension Reduction

Prediction of subcellular localization for proteases was performed using CELLO v2.5 [92] and PSORTb v3.0.2 [93]. Putative signal peptides were predicted with SignalP 5.0 [94]. All predictions are consolidated and shown in Appendix A. Numerical cellular localization data per sequence were used as features for dimension reduction techniques. This list includes wall, membrane, extracellular and intracellular scores from PSORTb (four features) and CELLO (four features), and also export pathway scores SP(Sec/SPI), TAT(Tat/SPI), LIPO(Sec/SPII), OTHER, intracellular and signal peptide possibility from SignalP (six features). The resulting matrix consisted of n sequences by 14 features. The set of sequences of size n includes the functional keratinase dataset, the putative non-keratinase dataset, and the three-strain dataset. The raw data matrix was standard-normalized, and dimensional reduction analysis was performed using Principal Component Analysis (PCA) and t-Distributed Stochastic Neighbor Embedding (t-SNE) (perplexity 30, 1000 iterations, fixed seed) [95]. Output coordinates for both methods were normalized using a min-max range. For the PCA, loadings for each feature were calculated from eigenvectors and depicted using arrows. The whole procedure was done using the Scikit-learn v.0.22.2 Python package [96].

### 4.10. Clustering and Phylogeny of t-SNE Groups

t-SNE points, i.e., coordinates by sequence, were clustered into groups, called “t-SNE groups”, using the DBSCAN algorithm [49] implemented in Scikit-learn v.0.22.2. Multiple sequence alignment (MSA) for each t-SNE group was performed using MAFFT v7.455 [97] with options G-ins and -maxiter 1000. Average occupancy (average number of residues per position in the alignment) and MSA filtering by occupancy were obtained using the Prody Python package [50]. Positions with occupancy below 70% were removed from original MSAs to generate more compact MSAs. A substitution model was fitted to each MSAs using ProtTest 3 [98], considering all distributions plus I and G models, and then a multithread maximum likelihood tree was obtained using RaxML v.8.2.12 [99] with the best parameters calculated by ProtTest and the rapid bootstrapping configuration (-f a option, 1000 bootstraps). Because no outgroup sequence was provided, the trees were midpoint rooted. Ancestral state reconstruction on trees [100] over the discrete categories: Keratinase, non-keratinase, three-strain, and keratinase-linked protein was performed using the package Phytools in R [101]. The set up was defined in 1000 iterations and an “ER” model, which means that equal rates for all permitted transitions.

## 5. Conclusions

Our robust comparative genomic analysis between three *Streptomyces* strains with varying keratinolytic activities permitted the identification of a set of putative proteases that could potentially be involved in the keratinolytic capacity of *Streptomyces* sp. G11C. According to peptidase family classification and orthogroup identification, we consider as promising candidates: (1) Unique putative peptidases in the keratinolytic strain G11C (17 unassigned p-orthogroup peptidases), including those belonging to the peptidase families C02 and S53; and (2) three peptidases present in the orthologous groups shared between strains G11C and CHD11, but not present in the non-keratinolytic strain Vc74B-19. Additionally, similarity network analysis identified three communities of keratinases-linked peptidases belonging to families S01, S08, and M04. Complementing this information with sub-cellular localization data and phylogenetic analysis, we identified seven promising genes likely to encode potential keratinases from *Streptomyces* sp. G11C, belonging to peptidase families S01 and S08. These findings provide genetic information for the proteomic analysis in the keratinolytic strain G11C, described in related work [40], which functionally validates the predictions accomplished in this study. This is the first comprehensive bioinformatics analysis that complements comparative genomics with phylogeny, network similarities, and cellular localization prediction to provide a set of genes considered to encode putative keratinases. This semi-supervised pipeline, involving t-SNE clustering on cellular localization data, is a novel approach in the keratinase literature, and we consider it as a significant advance that will help build more sophisticated pipelines in the future. In addition, it can be useful for various other hydrolytic enzyme families, such as lipases, glycosidases, esterases, among others. 

## Figures and Tables

**Figure 1 marinedrugs-19-00286-f001:**
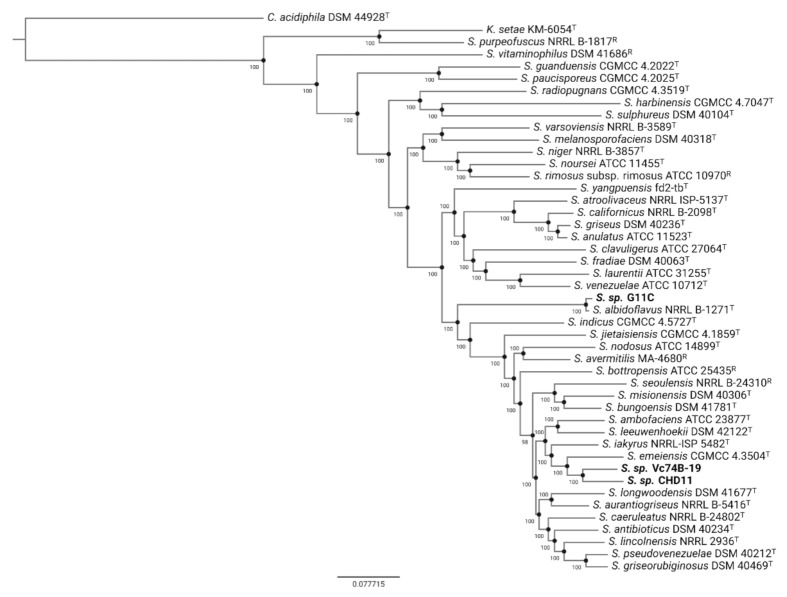
Phylogenomic tree of selected *Streptomyces* strains using 634 identified single-copy orthogroups via Orthofinder and subsequent Fasttree inference. Genomes of type (T) and reference (R) strains belonging to the *Streptomyces* genus were retrieved from the PATRIC database. *Catenulispora acidiphila* DSM44928 and *Kitasatospora setae* KM-6054 were used as outgroups, rooting the tree at the *Catenulispora* node. Bootstrap support values (b = 1000) are depicted for each branch. Strains sequenced in this study are displayed in bold font. The following abbreviations apply: *S.*, *Streptomyces*; *K.*, *Kitasatospora*; *C.*, *Catenulispora*.

**Figure 2 marinedrugs-19-00286-f002:**
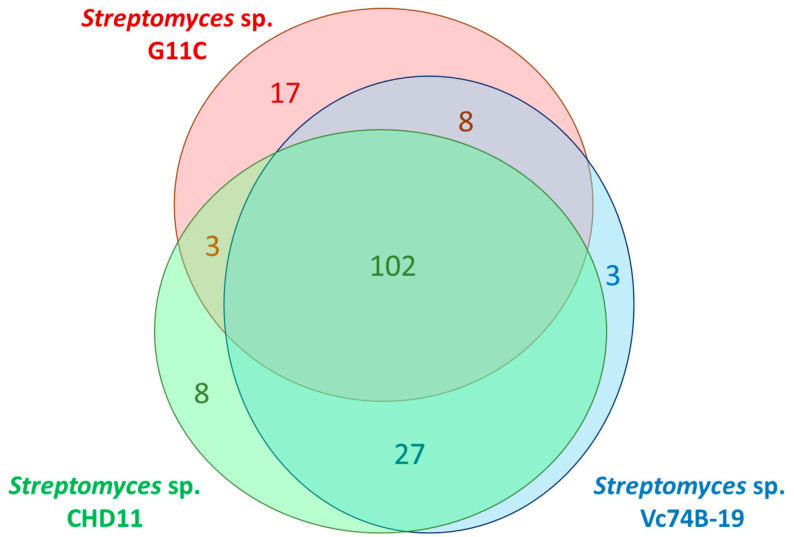
Venn diagram of common p-orthogroup representatives between streptomycete strains G11C, CHD11, and Vc74B-19. Numbers indicate the number of p-orthogroups found for each strain or between strains.

**Figure 3 marinedrugs-19-00286-f003:**
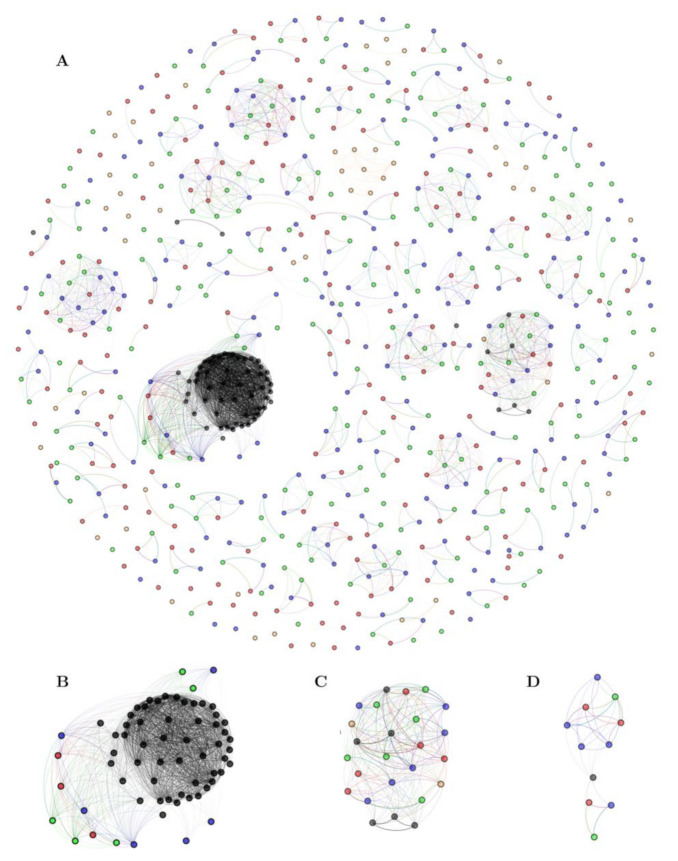
Protease similarity network, including the three-strain (584 sequences), keratinase (61 sequences), and non-keratinase (50 sequences) datasets. The E-value threshold of the blast alignment for the network is 1 × 10^−40^ Each node represents an identified putative protease, and the color fill indicates the origin of the sequence: Red, strain G11C; green, strain CHD11; blue, strain Vc74B-19; black, functionally known keratinases; yellow, putative non-keratinases. Edge transparency was adjusted to represent E-value difference: Darker edges correspond to smaller E-values. (**A**) entire network, (**B**–**D**) zoom into particular network clusters possessing known keratinases: Community 1, 4, and 41, respectively.

**Figure 4 marinedrugs-19-00286-f004:**
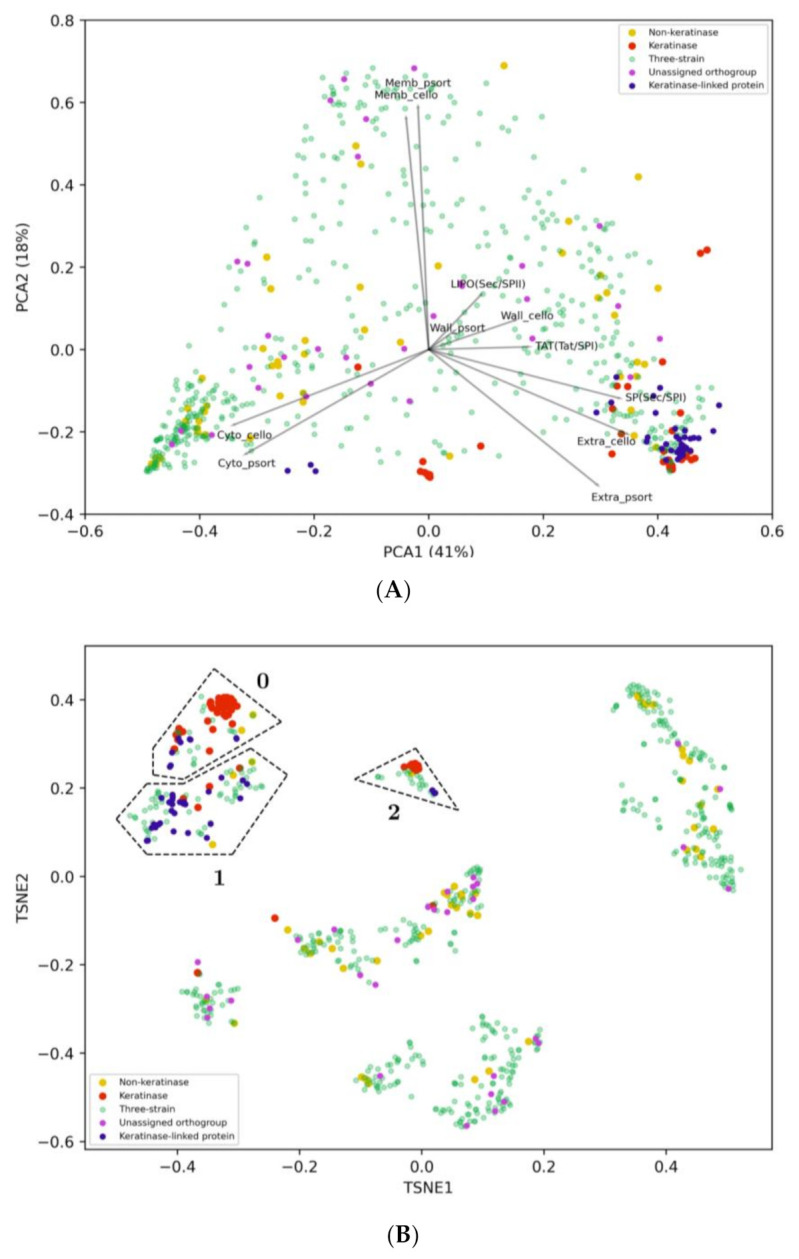
(**A**) Two-dimensional PCA using localization features for proteases. Protease sequences were retrieved from genomes of strain G11C, Vc74B-19, and CHD11. The x- and *y*-axis explain 41% and 18% of the observed variance of the data, respectively. Functional keratinases are depicted in red, putative non-keratinases in yellow, sequences from our three streptomycete genomes (i.e., three-strain category) in green, sequences without assigned p-orthogroup in magenta and keratinase-linked proteins, according to the network, are depicted in blue. Loadings represent the cellular localization features: Wall, membrane, cytoplasmic, and extracellular of the software CELLO and PSORTb, in addition to putative signal peptides: LIPO(Sec/SPII), SP(Sec/SPI), TAT(Tat/SPI) predicted with SignalP. (**B**) Two-dimensional t-SNE using localization features for the same group of proteases. Defined t-SNE group 0, 1, and 2 are enclosed by dashed polygons.

**Figure 5 marinedrugs-19-00286-f005:**
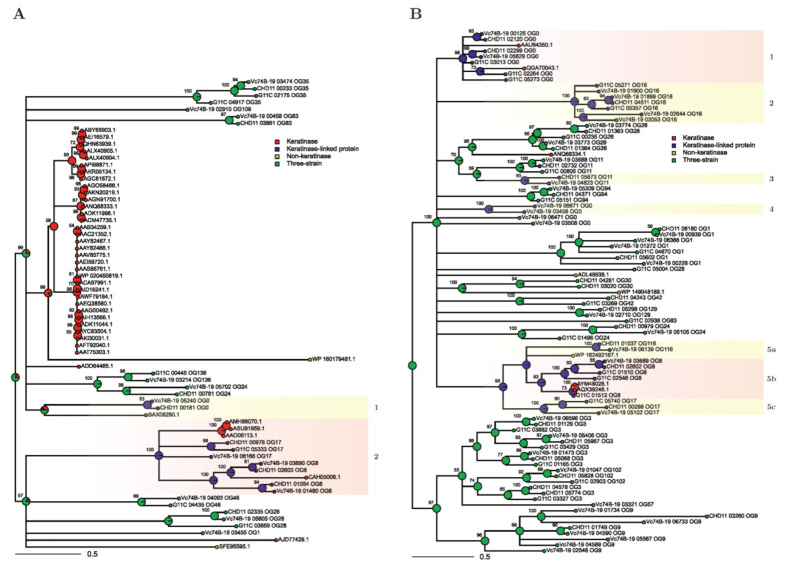
Maximum likelihood trees from filtered MSA of t-SNE group 0 (**A**) and t-SNE group 1 (**B**) sequences. Ancestral state probabilities inferred in internal nodes for each category (functional keratinase, keratinase-linked protein, three-strain, and non-keratinase) are depicted as pie charts, with a total of 1 for each pie chart. Support values based on bootstrapping are indicated for each node. Selected clades under stipulated criteria are enclosed by light red boxes, while discarded clades are enclosed by light yellow boxes.

**Figure 6 marinedrugs-19-00286-f006:**
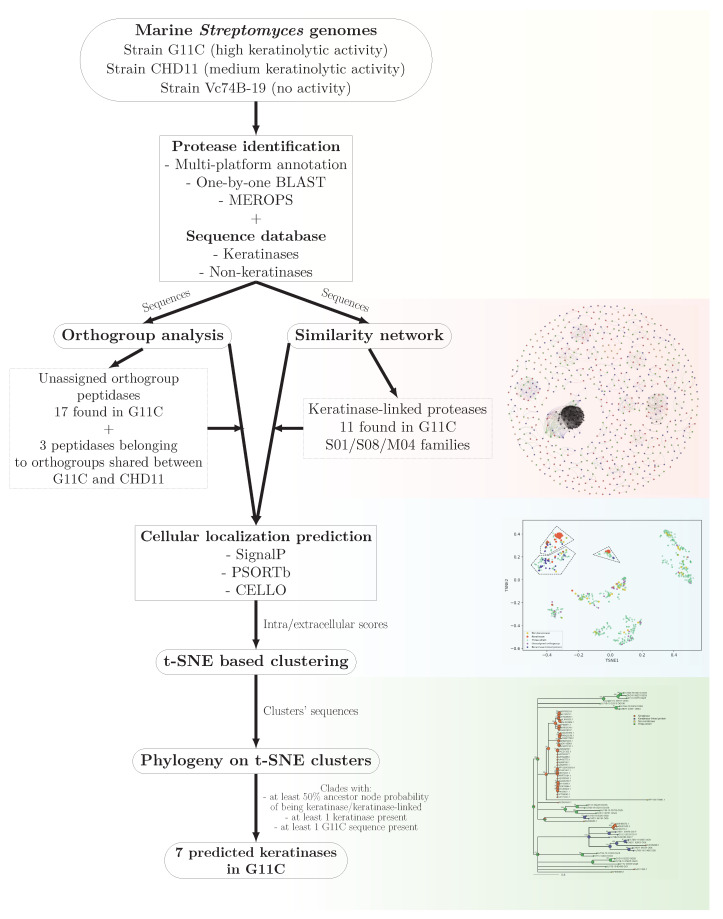
Bioinformatic pipeline to predict potential keratinases in *Streptomyces* sp. G11C. This analysis integrates a series of steps, including comparative genomics with network similarities, cellular localization prediction, and phylogeny to provide a set of genes considered to encode putative keratinases.

**Table 1 marinedrugs-19-00286-t001:** Assembly statistics of analyzed *Streptomyces* genomes.

Statistical Data	*Streptomyces* sp. G11C	*Streptomyces* sp. CHD11	*Streptomyces* sp. Vc74B-19
Total number of paired reads	7,253,694	9,168,658	8,356,602
Total raw reads bases (Gbp)	187.73	232.83	213.71
Assembly size (bp) (≥500 bp contigs)	6,873,298	7,469,836	7,625,040
Number of contigs (≥500 bp)	167	91	197
Contigs (N50) (kb)	204,147	98,291	76,195
G + C content (%)	73.15	71.67	72.33
Predicted CDS	5953	6661	6835
CheckM completeness (%)	99.53	100	99.24
L50 statistics	24	12	31

**Table 2 marinedrugs-19-00286-t002:** Characteristics of putative proteases genes in *Streptomyces* strains G11C, CHD11, and Vc74B-19.

Number of:	*Streptomyces* sp. G11C	*Streptomyces* sp. CHD11	*Streptomyces* sp. Vc74B-19
ORFs encoding putative proteases	179	198	207
Putative proteases classified into a peptidase family	151	166	177
p-orthogroups in each strain	113	132	137
“unassigned p-orthogroup” peptidases	17	8	3

**Table 3 marinedrugs-19-00286-t003:** Peptidase sequence distribution in t-SNE groups.

	Number of Peptidases Present in:		
Category	t-SNE Group 0	t-SNE Group 1	t-SNE Group 2
Functional keratinase	40	5	10
Keratinase-linked sequence—strain G11C	1	9	1
Keratinase-linked sequence—strain CHD11	4	8	1
Keratinase-linked sequence—strain Vc74B-19	4	11	1
three-strain category—strain G11C	5	13	6
three-strain category—strain CHD11	4	18	5
three-strain category—strain Vc74B-19	8	24	7
Putative non-keratinase	2	3	1

## Data Availability

Complete genome sequences of *Streptomyces* sp. G11C, CHD11, and Vc74B-19 are available in NCBI Genbank under WGS accession numbers JABTTT000000000, JABTTS000000000, and JABTTR000000000, respectively. Prokka annotations of the genomes of *Streptomyces* sp. G11C, CHD11, Vc74B-19 are available via Figshare through the following link https://doi.org/10.6084/m9.figshare.13133270.v1 (accessed on 28 October 2020). Fasta headers are consistent with reported sequence labels in this work.

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
