# Peer review of "An Integrative Bioinformatic Analysis for Keratinase Detection in Marine-Derived Streptomyces"

_marinedrugs, 2021, doi:10.3390/md19060286_

Round 1

Reviewer 1 Report

The study by Valencia et al, proposes a bioinformatic pipeline to study the genomes of three Streptomyces strains with the aim of identifying genes responsible for the high keratinolytic activity in one of the strains. In contrast, the other two showed low or not activity according to previous studies from the same group.

The main problem of this study is that they do not establish properly a pipeline. Rather, a series of bioinformatic tests have been applied with no obvious connection between them. Moreover, neither in the discussion nor in the conclusions, the results obtained with the different tests are compared. It is not clearly stated in the manuscript if the same genes were detected after applying the different tests. The manuscript should be improved by indicating clearly how the different test complement each other and how do they compare in terms of results. A graphical representation of the pipeline could be helpful

Major comments:

1. It is unclear how the results of the study are validated by previous results of the group. The authors claim (lines 82-83) that their results are validated because 18 out of 24 of the predicted proteases are present in the secretome described in the previous study (lines 467 and 468). Later, in lines 397-9 it is indicated that “strain G11C presented more “unassigned p-orthogroup” peptidases, of which 12 were actively involved in keratin degradation” referring to the same previous study of the group. It is not indicated if they are referring to the same proteins.

2. The section on comparative genomics is not properly described. The introduction to the section on peptidase identification (lines 128-138) and material and methods should be rewritten. For instance, it is not mentioned in the results section that the first approach for protease/peptidase detection was based in the genome annotation as shown in Table S4. The discussion on the possible role of the unique groups detected in the MEROPS families and orthogroups could be simplified since the peptidases in each orthogroup detected always belongs to the same family. Moreover,he authors should give a clear criterion to define the interest of the groups detected. Sometimes they mention as potential candidates the peptidases orthogroups shared by strains G11C and CHD11 (lines 166-170) or the peptidases unique to strain G11C (lines 180- 183). Also, in lines 188-200 the authors discuss the possible interest of the study of families s01, S08 and m04 for detecting keratinolytic enzymes, ”in spite of belonging to  orthogroups shared by the three strains analyzed” (lines 192-196)

3. The text is difficult to read and wording is unprecise in some cases. In lines 624-27 the authors indicate that the strain G11C posses some candidates: from “families evidenced as unique in the keratinolytic strain G11C (17 unassigned p-orthogroup peptidases), including peptidase families C02 and S53”. In my opinion it should have been indicated that there is only protein detected in each of those two last families as indicated in lines 147-9

4. Which was the criterion for selecting only some t-SNE groups to perform the phylogenetic analysis?. In general, the results of this part are difficult to follow and, again, they lack connection with the previous sections.

5. The authors describe three network communities involving possible keratinolytic enzymes. Networks 1 and 4 seems to be evident, but network 41 contains a single keratinase “control” enzyme which seems to be loosely connected with the others (Fig. 3D). Some of the peptidases previously mentioned as of possible interest, for example the unassigned to any orthogroups unique to G11C don’t below to any community. This should have been discussed.

6. The results on cellular location are not properly integrated in the study. In lines 271-3 it is mentioned that most of unassigned to any orthogroups are cytoplasmic. Does this observation include the peptidases unique to G11C previously considered as candidates?

Minor comments:

  1. In table S1 the “closest type strains” to the strains used in this study are shown. It is unclear how was this determined. Surprisingly in the ANI studies the “closest type strains” to strains CHD11 and Vc74B-19 (Streptomyces aurantiogriseus NRRL B-5416T and Streptomyces albogriseolus NBRC 3413T, respectively) were not used. Any reason for that? Both should have been included to get a meaningful result of ANI experiments.
  2. The Veen diagram shown in Fig. 2 should be modified so the sizes of the circles and intersections could show better the number of proteases in each section.
  3. Streptomyces should be written in italics in all the text.

Author Response

The study by Valencia et al, proposes a bioinformatic pipeline to study the genomes of three Streptomyces strains with the aim of identifying genes responsible for the high keratinolytic activity in one of the strains. In contrast, the other two showed low or not activity according to previous studies from the same group.

The main problem of this study is that they do not establish properly a pipeline. Rather, a series of bioinformatic tests have been applied with no obvious connection between them. Moreover, neither in the discussion nor in the conclusions, the results obtained with the different tests are compared. It is not clearly stated in the manuscript if the same genes were detected after applying the different tests. The manuscript should be improved by indicating clearly how the different test complement each other and how do they compare in terms of results. A graphical representation of the pipeline (Richi) could be helpful

Answer: Thank you for all of your comments. Both the orthogroup and similarity network analyses are done separately and are categorized as groups of interest. These groups were then filtered with the following criteria: cellular localization, t-SNE clustering and phylogeny of t-SNE clusters. The complete list of promising sequences, analysed in each section can be found in Table S8.

To provide more clarity, an additional section has been added in the results (section 2.5, lines 387-430) that summarizes the steps and main results obtained in the bioinformatic analysis, including an additional figure (Figure 6) depicting a graphical representation of the pipeline we used.

Major comments:

  1. It is unclear how the results of the study are validated by previous results of the group. The authors claim (lines 82-83) that their results are validated because 18 out of 24 of the predicted proteases are present in the secretome described in the previous study (lines 467 and 468). Later, in lines 397-9 it is indicated that “strain G11C presented more “unassigned p-orthogroup” peptidases, of which 12 were actively involved in keratin degradation” referring to the same previous study of the group. It is not indicated if they are referring to the same proteins.

Answer: As mentioned by the reviewer, 18 of 24 proteases predicted in this bioinformatic analysis were found in the secretome of the strain G11C in our previous study, 6 of 7 potential keratinases predicted by phylogenetic analysis, and 12 of 17 unique genes of the G11C genome (“unassigned p-orthogroup” peptidases) identified by orthogroup classification. We provide additional information in the introduction (lines 82-84) and the discussion (lines 456-459), as suggested.

2.1. The section on comparative genomics is not properly described. The introduction to the section on peptidase identification (lines 128-138) and material and methods should be rewritten. For instance, it is not mentioned in the results section that the first approach for protease/peptidase detection was based in the genome annotation as shown in Table S4.

Answer: Considering your suggestion, additional information has been provided in the results section of comparative genomics, in lines 135,136, and 141. Also, sections 4.5 and 4.6 (lines 620-647) in materials and methods have been reordered to improve description.

2.2. The discussion on the possible role of the unique groups detected in the MEROPS families and orthogroups could be simplified since the peptidases in each orthogroup detected always belongs to the same family. Moreover, the authors should give a clear criterion to define the interest of the groups detected. Sometimes they mention as potential candidates the peptidases orthogroups shared by strains G11C and CHD11 (lines 166-170) or the peptidases unique to strain G11C (lines 180- 183). Also, in lines 188-200 the authors discuss the possible interest of the study of families s01, S08 and m04 for detecting keratinolytic enzymes, ”in spite of belonging to  orthogroups shared by the three strains analyzed” (lines 192-196)

Answer: Thank you for your comments. Regarding the classification of MEROPS peptidase families and orthogroup analysis, we consider that both analyses are complementary, because they provide valuable information on their own. For example, the first analysis allowed us to identify two peptidase families present only in the strain G11C (families C02 and S53, lines 152-153), whereas the orthogroup analysis allowed us to identify 17 peptidases that are not assigned to any shared orthogroup, also being unique to strain G11C. By complementing both analyses, we were able to assign each of these peptidases to different families (S01, S16, S15, S53, S51, M86, M50, M56, C82 and C02 and 7 unassigned peptidases to any family, see lines 185-188), including the families mentioned above. Therefore, we believe that a detailed discussion provides the reader with important information on the similarities and differences between the three strains, providing evidence on the factors that could be contributing to keratinolytic activity.

To improve the understanding of the manuscript and the criteria used, a new section 2.5 (lines 387-430) has been added. Also, to have a clearer understanding of the pipeline, a new figure 6 showing a graphical representation of the pipeline, as suggested by the reviewer.

  1. The text is difficult to read and wording is unprecise in some cases. In lines 624-27 the authors indicate that the strain G11C posses some candidates: from “families evidenced as unique in the keratinolytic strain G11C (17 unassigned p-orthogroup peptidases), including peptidase families C02 and S53”. In my opinion it should have been indicated that there is only protein detected in each of those two last families as indicated in lines 147-9

Answer: Thank you for the correction. Our apologies for this mistake, you are right. What we wanted to say is that we consider as interesting candidates the unique peptidases of strain G11C (mentioned in lines 185-188), where the peptidases belonging to the unique families of this strain (C02 and S53 families) are included. We correct this sentence according to the aforementioned, in lines 714-718.

  1. Which was the criterion for selecting only some t-SNE groups to perform the phylogenetic analysis?. In general, the results of this part are difficult to follow and, again, they lack connection with the previous sections.

Answer: The specific aim of the PCA and t-SNE analyses was to identify groups of sequences with extracellular localization characteristics. Due to the molecular structure of keratin, the production of extracellular keratinases is required. Therefore, to identify those promising extracellular sequences, the information obtained from the similarity network and orthogroups identification analyses (keratinase-linked proteases, unassigned p-orthogroup sequences, and the remaining into the three-strain category) was incorporated in the PCA and t-SNE analyses. Thus, the clustering of the sequences using the t-SNE analysis allowed the identification of three well-defined t-SNE groups 0, 1 and 2, with sequences predicted as extracellular, and which contained a high number of functional keratinases and proteases of the three strains linked to keratinases. Therefore, these three groups were selected for containing possible candidates encoding potential extracellular keratinases. Subsequently, this information was incorporated into a phylogenetic analysis, to identify the sequences closest to functional keratinases, and thus identify the potential keratinases of strain G11C. To emphasize the purpose of this analysis, information was added on lines 265, 303, and 305-307.

  1. The authors describe three network communities involving possible keratinolytic enzymes. Networks 1 and 4 seems to be evident, but network 41 contains a single keratinase “control” enzyme which seems to be loosely connected with the others (Fig. 3D). Some of the peptidases previously mentioned as of possible interest, for example the unassigned to any orthogroups unique to G11C don’t below to any community. This should have been discussed.

Answer: Thank you for your suggestion. More information related to communitiy 41 and unassigned proteases unique to G11C has been added to the discussion in lines 480 and 494.

  1. The results on cellular location are not properly integrated in the study. In lines 271-3 it is mentioned that most of unassigned to any orthogroups are cytoplasmic. Does this observation include the peptidases unique to G11C previously considered as candidates?

Answer: As indicated by the reviewer, the group of unassigned p-orthogroup sequences includes the unique sequences of Streptomyces sp. G11C. This information was incorporated in lines 279-281. Subsequently, discussion concerning this fact has been improved in lines 456-464.

Minor comments:

  1. In table S1 the “closest type strains” to the strains used in this study are shown. It is unclear how was this determined. Surprisingly in the ANI studies the “closest type strains” to strains CHD11 and Vc74B-19 (Streptomyces aurantiogriseus NRRL B-5416T and Streptomyces albogriseolus NBRC 3413T, respectively) were not used. Any reason for that? Both should have been included to get a meaningful result of ANI experiments.

Answer: Thank you for your comment. The “closest type strains” informed in Table S1 was previously carried out by Cumsille et al. (2017) and is based on the 16S rRNA gene sequence similarity. Although this paper was cited in Table S1, additional information in the corresponding column of this table has been given: “Closest Type Strain by 16S rRNA gene sequence similarity (Cumsille et al, 2017)”.

Regarding the two genomes mentioned to add in the analysis, in the case of S. albogriseoulus NBRC 3413T, we initially considered it, however the genome assemblies are tagged as an “anomalous assembly” in the NCBI Genbank (https://www.ncbi.nlm.nih.gov/assembly/?term=txid1887[Organism:noexp]), so we decided to leave it out of our analysis. Concerning the S. aurantiogriseus NRRL B-5416 genome, a new phylogenomic tree and an ANIm matrix to include this strain. The new figures 1 and S1 are presented in the manuscript. These criteria were also added to the Methods section for clarity (see lines 606-610).

  1. The Veen diagram shown in Fig. 2 should be modified so the sizes of the circles and intersections could show better the number of proteases in each section.

Answer: Figure 2 has been changed according to the reviewer´s suggestion.

  1. Streptomyces should be written in italics in all the text.

Answer: We agree with the reviewer and have made the changes accordingly.

Reviewer 2 Report

In the manuscript, “An Integrative Bioinformatic Analysis for Keratinase Detection in Marine-derived Streptomyces” the authors describe bioinformatic methodology to assess likely keratinases present in the genomes of keratinolytic Streptomyces. The manuscript is very well written and the methods are appropriately described. Upon reading, it was a bit jarring to realize that the confirmed proteases present in the secretome data from the authors was not part of this manuscript and had previously been reported. Of course a complete study with this data leading into such confirmation would have been ideal, but I think that the authors provide a sufficient bioinformatic process to explore likely kerantinases for a standalone manuscript. To address this, the authors should improve the section of the introduction (line 82) to clarify that the secreted keratinases were validated in a previous study. I suspect that this might result in concern that the authors preemptively knew the likely proteases and targeted them during their bioinformatic analysis, but I do not suspect this whatsoever. The bioinformatic analysis is very thorough; I especially enjoyed the correlation between the similarity network analysis and cellular localization data. This is a great combination for this specific class of enzymes, and I suspect readers will appreciate the described methodology. The authors do a nice job presenting this exact use case in their conclusions, and I wholeheartedly agree. Eventual confirmation of the involvement of their identified enzymes in the keratinolytic activity of Streptomyces sp. strain G11C using traditional genetic techniques would be an excellent subsequent study, but I do not think it is necessary to add these experiments to this current manuscript.

“Streptomyces” and species designations should be italicized throughout the manuscript.

Table 2: “p-ortogroups” should be corrected to “p-orthogroups”

line 223: “e-value” should be correct to “E-value” for consistency

Strain identifiers should be consistent throughout. ie. Streptomyces sp. strain G11C is referenced 3 different ways in the conclusions section of the manuscript.

Author Response

In the manuscript, “An Integrative Bioinformatic Analysis for Keratinase Detection in Marine-derived Streptomyces” the authors describe bioinformatic methodology to assess likely keratinases present in the genomes of keratinolytic Streptomyces. The manuscript is very well written and the methods are appropriately described. Upon reading, it was a bit jarring to realize that the confirmed proteases present in the secretome data from the authors was not part of this manuscript and had previously been reported. Of course a complete study with this data leading into such confirmation would have been ideal, but I think that the authors provide a sufficient bioinformatic process to explore likely kerantinases for a standalone manuscript. To address this, the authors should improve the section of the introduction (line 82) to clarify that the secreted keratinases were validated in a previous study. I suspect that this might result in concern that the authors preemptively knew the likely proteases and targeted them during their bioinformatic analysis, but I do not suspect this whatsoever. The bioinformatic analysis is very thorough; I especially enjoyed the correlation between the similarity network analysis and cellular localization data. This is a great combination for this specific class of enzymes, and I suspect readers will appreciate the described methodology. The authors do a nice job presenting this exact use case in their conclusions, and I wholeheartedly agree. Eventual confirmation of the involvement of their identified enzymes in the keratinolytic activity of Streptomyces sp. strain G11C using traditional genetic techniques would be an excellent subsequent study, but I do not think it is necessary to add these experiments to this current manuscript.

Answer: Thank you very much for your comments concerning our manuscript. This is especially encouraging for my two students that are co-first authors and have done a tremendous job.

The initial idea was to publish this paper first and then the second paper with the proteomics confirmation, however it didn´t work out that way. The bioinformatic analysis was done separately and we were very pleased to confirm prediction with our proteomics data. Once again, thank you.

“Streptomyces” and species designations should be italicized throughout the manuscript.

Answer: We agree with the reviewer and changed throughout the manuscript.

Table 2: “p-ortogroups” should be corrected to “p-orthogroups”

Answer: Has been changed as suggested in Table 2.

line 223: “e-value” should be correct to “E-value” for consistency

Answer: Has been changed as suggested.

Strain identifiers should be consistent throughout. ie. Streptomyces sp. strain G11C is referenced 3 different ways in the conclusions section of the manuscript.

Answer: We agree with the comment and changed accordingly.

Reviewer 3 Report

The article by Ricardo Valencia et al. investigates the first multi-step bioinformatics analysis that complement comparative genomics with phylogeny and cellular localization prediction, for the prediction of genes encoding putative keratinases in streptomycetes.

The title reflects the article’s content.

The results and observations are relevant to the problem posed and provide proof for the consistency of literature data regarding the keratinases, keratinolytic proteases, marine-derived Streptomyces, and multi-step bioinformatics. 

The work is original and scientifically reliable since the methods are appropriate and adequately described.

The article is very well presented with proper use of tables and figures.

The speculations made are reasonable and all the interpretations are warranted by the data gathered.

The message is transmitted clearly: the comparative genomic analysis between three Streptomyces strains with varying keratinolytic activities permits the identification of a set of putative proteases that could potentially be involved in the keratinolytic capacity of Streptomyces sp. G11C.

The references are up to date and relevant.

The overall design of the study is good, the undertaken research is properly described and the conditions are well defined.

The English is clear and it is understandable to the reader.

I recommend publishing the article.

Author Response

The article by Ricardo Valencia et al. investigates the first multi-step bioinformatics analysis that complement comparative genomics with phylogeny and cellular localization prediction, for the prediction of genes encoding putative keratinases in streptomycetes.

The title reflects the article’s content.

The results and observations are relevant to the problem posed and provide proof for the consistency of literature data regarding the keratinases, keratinolytic proteases, marine-derived Streptomyces, and multi-step bioinformatics. 

The work is original and scientifically reliable since the methods are appropriate and adequately described.

The article is very well presented with proper use of tables and figures.

The speculations made are reasonable and all the interpretations are warranted by the data gathered.

The message is transmitted clearly: the comparative genomic analysis between three Streptomyces strains with varying keratinolytic activities permits the identification of a set of putative proteases that could potentially be involved in the keratinolytic capacity of Streptomyces sp. G11C.

The references are up to date and relevant.

The overall design of the study is good, the undertaken research is properly described and the conditions are well defined.

The English is clear and it is understandable to the reader.

I recommend publishing the article.

Answer: We thank you very much for your comments concerning our manuscript. It was really nice, especially for the co-first authors to see that their work was very much appreciated. 

Round 2

Reviewer 1 Report

The authors have done a good job improving the manuscript and answering to the comments in the revision of the first submission.